# Probabilistic Human-like Gesture Synthesis from Speech using GRU-based WGAN

BOWEN WU, Osaka University, Japan

CHAORAN LIU, ATR, Japan

CARLOS T. ISHI, Guardian Robot Project, RIKEN, Japan

HIROSHI ISHIGURO, Osaka University, Japan

Gestures are crucial for increasing the human-likeness of agents and robots to achieve smoother interactions with humans. The realization of an effective system to model human gestures, which are matched with the speech utterances, is necessary to be embedded in these agents. In this work, we propose a GRU-based autoregressive generation model for gesture generation, which is trained with a CNN-based discriminator in an adversarial manner using a WGAN-based learning algorithm. The model is trained to output the rotation angles of the joints in the upper body, and implemented to animate a CG avatar. The motions synthesized by the proposed system are evaluated via an objective measure and a subjective experiment, showing that the proposed model outperforms a baseline model which is trained by a state-of-the-art GAN-based algorithm, using the same dataset. This result reveals that it is essential to develop a stable and robust learning algorithm for training gesture generation models. Our code can be found in https://github.com/wubowen416/gesture-generation.

CCS Concepts: • **Computer systems organization** → **Embedded systems**; *Redundancy*; Robotics; • **Networks** → Network reliability.

Additional Key Words and Phrases: gesture generation, social robots, generative model, neural network, deep learning

**ACM Reference Format:**
Bowen Wu, Chaoran Liu, Carlos T. Ishi, and Hiroshi Ishiguro. 2018. Probabilistic Human-like Gesture Synthesis from Speech using GRU-based WGAN. In *Woodstock '18: ACM Symposium on Neural Gaze Detection, June 03–05, 2018, Woodstock, NY*. ACM, New York, NY, USA, 16 pages. https://doi.org/10.1145/1122445.1122456

## 1 INTRODUCTION

Human-like agents can play various roles in this society, such as guidance, receptionist, TV hosts, presenter, and so on. These agents have human appearances, so that they are expected to behave like humans. During human interaction, gesture is one crucial non-verbal behavior that may convey different semantic as well as affective information. Thus, a system for generating natural gestures is necessary for an embodied or virtual human-like agent to fill the gap with humans.

There are various ways for generating gestures. For specific expressions, it can be hand-crafted by creating the movements on an agent by directly editing the control commands [3]. Mapping human motion data with word concepts through gesture functions and dialogue act information is also possible [13][14][19]. However, these methods need expert's knowledge and require extensive analysis of data. On the other hand, automatically creating generation models

using learning algorithms is straightforward, if a large-scale data is available. In this work, we aim at developing learning algorithms to train a model that can generate natural gestures, based on large-scale data.

Numerous deep learning-based models have been proposed to model human gestures. Even though the probabilistic models (i.e., models that can produce multiple outputs for one input) outperform deterministic models (i.e., models that produce single output for one input), the naturalness of the generated motion is still not achieving the human level [24]. Thus, in this work, we aim on building a better model for the generation of gestures. We proposed a novel deep-learning-based model for gesture generation and successfully trained it using the loss function we designed. We designed a complete synthesis protocol for gesture generation, which outputs the rotation vector of the upper body joints. We confirmed that the synthesized motions of the proposed model outperform the baseline model via objective and subjective measures.

The rest of this article is organized as follows. In section II, we present the studies related to the present work. Section III provides a complete explanation of the proposed system. In section IV, the experiment for evaluating the proposed system is detailed. In section V, the obtained results are discussed.

## 2 RELATED WORK

### 2.1 Speech-Driven Gesture Synthesis via Learning

Recently, deep learning methods have been widely used on the gesture generation task because several open-access human gesture databases have been developed in these years [23][7]. Deterministic models assume that there is only one gesture corresponding to one segment of audio. For example, [11] used LSTM (long short term memory) to realize the mapping from MFCC (Mel-Frequency Cepstral Coefficient) features extracted from the input audio to gestures. [16][17] analyzed how different features of audio input affect the result, and proposed to use low-dimensional manifold as training target instead of high-dimensional raw data. A style transfer model was developed to generate gestures with personal trait using arbitrary audio input (i.e., different person's voice) [2]. Text of the speech was also included in the inputs to the model to generate gestures [18][25][1][6]. However, these models are trained using mean squared error (MSE) as loss function so that they can only produce one motion sequence for a given audio segment input.

On the other hand, since there can be multiple solutions of gesture sequence for one audio segment, probabilistic modeling of human gestures has been proposed. These models can produce different available motion sequences for one audio segment input. One approach is by using Glow-based model, which maps the source distribution to the target distribution [15]. For instance, MoGlow was used to model the distribution of human gestures [4]. In this model, one audio segment can be used to generate multiple motion sequences because the loss function for training MoGlow is the likelihood of the rotation values of each joint in the real data distribution. Another direction is to utilize generative adversarial network (GAN) [26][24]. The probabilistic generation is realized by inputting a randomly sampled noise vector to the generator. Different noise vectors can lead the model to produce different motions. A common failure, called mode collapse, in GAN training can be reduced by using the learning algorithm of unrolled-GAN[24]. However, Glow-based models have a huge number of parameters to learn, which is known as being hard to train. The generated motions of the previously mentioned GAN model appears to move too much, which negatively affects the naturalness. The reason is likely to be that the model only partially covers the real motion distribution, failing to learn the part of the real motion distribution with less or no movements. In other words, due to the learning algorithm of the model, mode collapse is still happening and severely damages the performance of the model.

## 2.2 WGAN-GP

Although vanilla-GAN has a common issue that the model does not converge so that the training is unstable and it is hard to diagnose in order to improve the performance, the learning algorithm of Wasserstein-GAN (WGAN) is supposed to be effective for training GAN model, because it is mathematically convergent, and thus provides a reliable indication of the progress of the training process [5]. WGAN has shown its success in multiple research areas including speech synthesis[27], music generation[20], and natural language modeling[21].

Gradient penalty is an advanced technique to stabilize the training process of GAN. It penalizes the norm of the gradients on the parameters of the discriminator model to ensure Lipschitz continuity [10].

In this work, we adopted the learning algorithm of WGAN-GP (WGAN with gradient penalty), to train a motion generator for the modeling of human gesture. The consideration is that with an advanced training algorithm to reduce the effect of mode collapse, the model can approximate the real motion distribution much better, thus generating more natural and human-like gestures.

## 3 PROPOSED GESTURE SYNTHESIS SYSTEM

In the proposed system, we formulate the gesture generation problem as modeling conditional distribution, given the observed speech. Specifically, given parameters extracted from a segment of speech, we try to sample corresponding gesture of same period from the conditional distribution modeled by a deep neural network. In addition, we employ Huber cost during the training process to explicitly reduce the discontinuity within the sampled gestures.

### 3.1 Data Pre-processing

Prosodic features are extracted from the speech signal to be used as conditional parameters for the GAN model. Specifically, voice pitch and power values were estimated each 10ms [12]. For the voice pitch features, the F0 (fundamental frequency) values were estimated by a conventional autocorrelation-based method. All the estimated F0 values were then converted to a musical (log) scale in semitone intervals before any subsequent processing. The power values were computed in dB units.

Although the motion data used in this work includes joints both in the upper body and lower body, which is recorded by Motion Capture Toolkit [23], only the upper body joints were used as target of modeling, since most movements in the dataset are concentrated on the upper body. As a result, the data of 12 joints are selected to form the training data.

### 3.2 Gesture Generator Architecture

An overview of the proposed system is shown in Fig. 1. First, the prosodic feature vector is extracted from the audio segment. Then, it is concatenated with a randomly sampled noise vector and fed into the generator. Through computation, the generator outputs frames of rotation vectors. The discriminator will be discussed in the following sub-section. Additionally, to avoid jerky motion, a low-pass filter is used to post-process the generated rotation vectors.

The generator consists of a prosodic feature extractor and a 2-layer bi-GRU (bi-directional gated recurrent unit) network. The bi-GRU network takes its input as the concatenation of the extracted audio features, the randomly sampled noise vector, and seed poses. The audio is segmented into 1.5 seconds with an overlap of 0.2 seconds. For each audio segment, 34 frames of motion are produced. Overlap of motion is interpolated using 4 frames at the last of the previous motion chunk, and 4 frames at the beginning of the next motion chunk. The discriminator composes a 1-dimensional

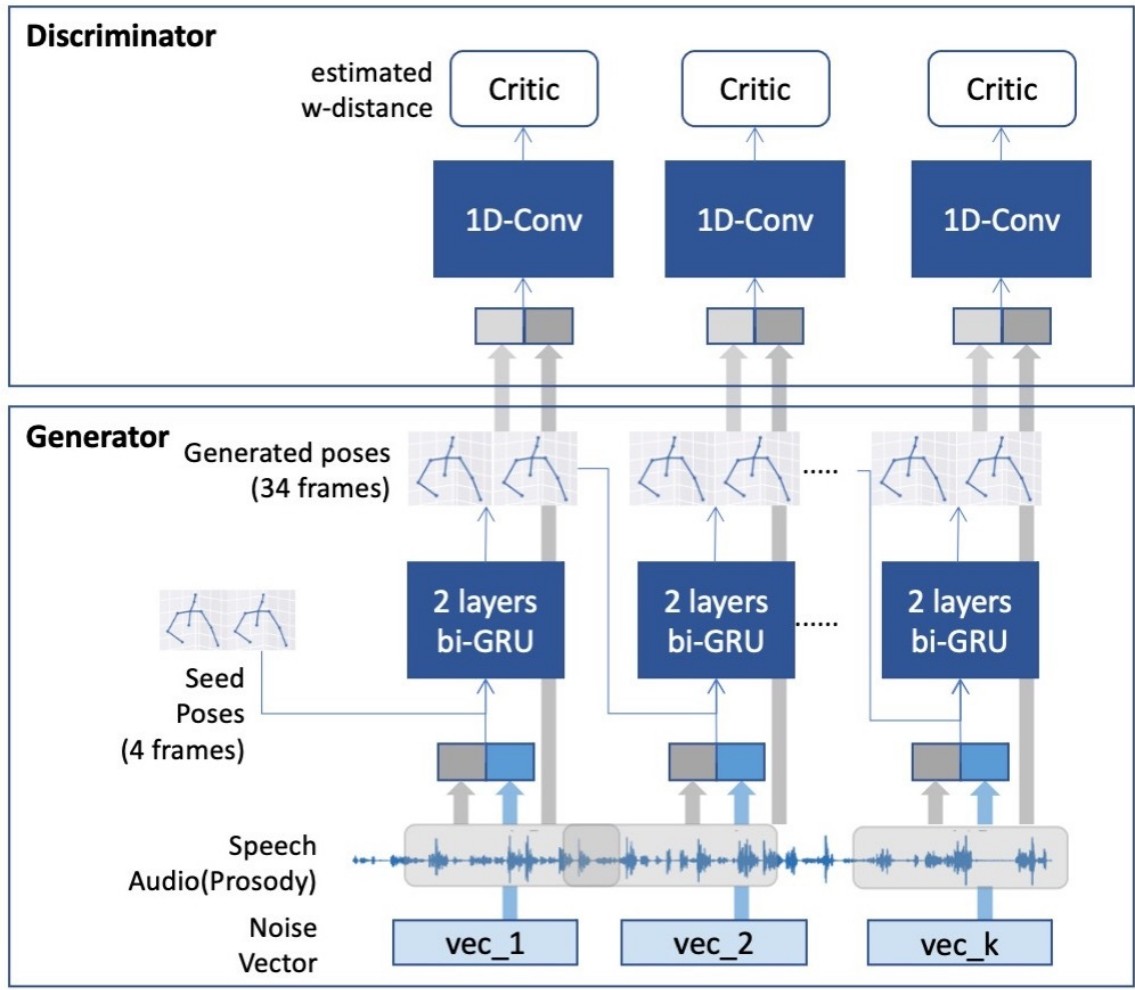

Fig. 1. An overview of the proposed system.

convolution layer, which takes input as the concatenation of the motion chunk and audio segment and outputs a scalar to indicate the distance between the real data distribution and the generated distribution.

### 3.3 Training

First of all, in order to realize a probabilistic generation, a discriminator is used to train the generator by an adversarial training. Secondly, the motion data in the dataset are re-shaped to short chunks (1.5 seconds for one chunk). This is by considering that a long sequence of motion can be divided into several short motion chunks. If the motions in these chunks are natural, and the transition between these chunks are also natural, the whole sequence of motion will be natural enough. This way, we can focus on training the generator to generate each short motion chunk and to realize the transition between these chunks, instead of producing a whole sequence of motions, which is more difficult

for learning. This brings us to use a convolutional layer based discriminator, as the length of each training sample is relatively short. Additionally, the input to the discriminator includes the extracted audio features to force the generated motion to be synchronized with the input audio. As a result, the loss for training the generator consists of three parts: the critic loss provided by the discriminator, the continuity loss for the transition between motion chunks, and the gradient penalty loss for training stability.

$$\mathcal{L}_{total} = \mathcal{L}_{critic} + \lambda_{gp} * \mathcal{L}_{gp} + \lambda_c * \mathcal{L}_{continuity} \tag{1}$$

*3.3.1 Critic Loss.* Critic loss is the traditional loss function of WGAN [5]. The critic loss is computed by an optimal discriminator which outputs the distance between the conditional distribution of the generated rotation vector and that of the real rotation vector, conditioned on the input audio features. The optimal discriminator is approximately obtained by previously training the discriminator to maximize the distance between the two conditional distributions. Then, the calculated distance is used as the loss for training the generator through back-propagation. The critic loss is defined as

$$\max_D \min_G \frac{1}{m} \sum_{i=1}^{m} D(y^{(i)}, s^{(i)}) - D(G(z, s^{(i)}), s^{(i)}) \tag{2}$$

where $y$ represents rotation vector, $s$ represents audio features, $z$ represents random sampled vector, $D$ is the discriminator, $G$ is the generator.

*3.3.2 Gradient Penalty (GP).* Gradient penalty is a regularization term for achieving stable training [10]. It punishes the norm of the gradients on the discriminator to be equal to 1, which requires to calculate the derivatives of the output of the discriminator with respect to its inputs. We applied the learning algorithm of WGAN-GP, in order to achieve a more stable training for our gesture generation model. Since our conditional discriminator has double inputs, we compared the result of only punishing the norm of the gradients based on one of them (only the part connected with the generator), and all of them. It turned out that there was no significant difference. Thus, under the current training setting, punishing only one side seems enough and time-saving. The gradient penalty loss is defined as

$$\mathcal{L}_{gp} = \frac{1}{m} \sum_{i=1}^{m} (\|\nabla_D^{(i)}\|_{L2} - 1)^2 \tag{3}$$

*3.3.3 Continuity Loss.* Since the generator can only generate motion chunks whose length is 1.5 seconds due to the training settings, continuity loss is used to force the frames at the beginning of the next motion chunk to be similar with those at the end of the previous motion chunk. This way, these motion chunks can be concatenated directly to form a whole motion sequence. The continuity loss is defined as

$$\mathcal{L}_{continuity} = \frac{1}{m} \sum_{i=1}^{m} \text{Huber}(y_{:k}^{(i)}, y_{-k:}^{(i-1)}) \tag{4}$$

where k is a hyper-parameter to define how many frames to force to be continuous with the previous motion chunk, and the Huber loss is defined in [8].

*3.3.4 Training procedure.* We train the proposed model on a Japanese dataset [23], in which the motion data is recorded with Motion Capture Toolkit. The data contains 1094 pairs of motion and audio. The total length of the data is 6 hours. To train the model, the learning rate was set to $10^{-4}$ for both the generator and discriminator. Batch size is set to

128. Lambda for continuity loss is set to 1. Lambda for GP is set to 10. The distribution for sampling noise vector is a Gaussian with 0 mean and one variance. The model was trained on RTX Titan GPU, for approximately 6 hours.

## 4 EVALUATION

It is not appropriate to evaluate motions using objective measures such as average position error for value-level comparison, because a motion can look natural even if the score is low based on these measures [25]. Kernel density estimation (KDE) is reported to be an appropriate measure for evaluating distributions as it outputs the log-likelihood of one distribution in another distribution, as used in [22][9]. Thus, we use KDE to objectively evaluate the generated results of the proposed model. Another common method for evaluating motions is based on user-study, which utilizes human perception. We conducted a user-study to assess the performance of the proposed model. The rotation values of joints is not straightforward for humans, but they can be visualized through motions in an avatar. In this work, we implemented motions on a virtual avatar using the Unity software (Fig. 2).

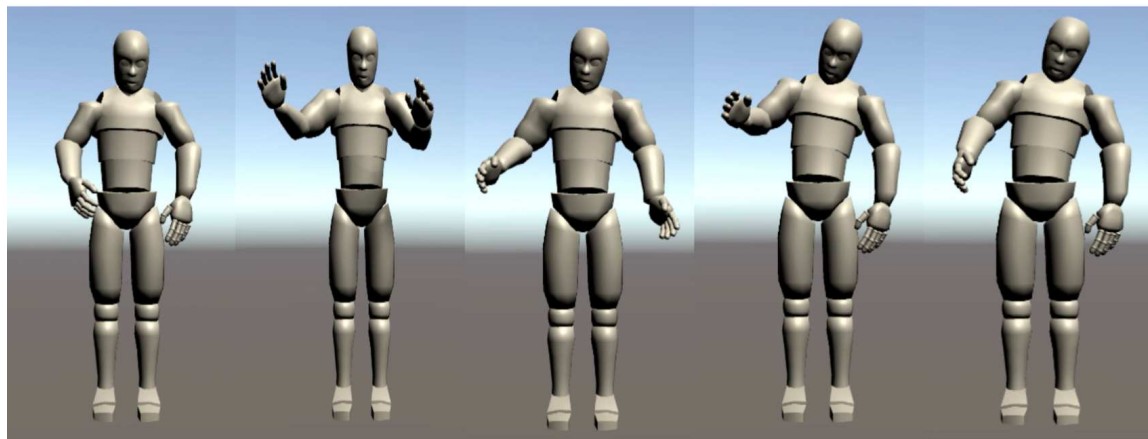

Fig. 2. Snapshots of upper-body motions synthesized in the avatar.

We compare the generated motion of the proposed model with different control groups using video clips. The details are shown below:

- **Ground Truth (GT)**. Real data recorded from human using Motion Capture Toolkit. Before applied to the avatar, a low-pass filter is used to pre-process (smooth) the data to avoid jerk motion, which may be due to the recording equipment.
- **Baseline**. The baseline model we chose is a state-of-the-art model for generating coordinate values of joints, trained on the same dataset used for training the proposed model. This model is a GAN-based probabilistic gesture generation model proposed in [24]. In order to make comparison, we trained a neural network to convert

the coordinate values to joint rotation values for the corresponding joints. The training data is the data used to train the proposed model.

- **Proposed Model**. As the goal is to realize probabilistic generation, the model is able to produce multiple motion sequences for one audio segment. To assess the result of different generated samples, we prepare two videos for the proposed model within each utterance set. Note that they are synthesized using different noise vectors for the same audio input.

## 4.1 Objective Evaluation

The motions generated using the proposed model and baseline model were used to fit a KDE model. The optimal bandwidth was obtained using a grid search with 3-fold cross-validation. Then, the log-likelihood of the real motion data in the test set was calculated using the fitted KDE model. Thus, as the output value tends to be larger, the motions better fit the real data distribution. A comparison between different models is shown in Table 1. The result of GT (Ground Truth) is calculated by computing the likelihood of the ground truth data using the KDE model fitted by the ground truth data itself. This can be seen as the best result that can be achieved. As the log-likelihood of a model tends to be similar with that of GT, the model will better fit the GT data distribution. It can be observed from the results that the proposed model achieves log-likelihood valued closer to the GT data.

Table 1. KDE evaluation results for different models.

| Model | Log-likelihood | Standard Error |
|---|---|---|
| GT | -122.53 | 0.23 |
| Baseline | -152.37 | 25.26 |
| Proposed | -125.60 | 0.10 |

## 4.2 Subjective Evaluation

There are three aspects that were measured for evaluating the proposed model, which are naturalness of the motion, the time consistency between the motion and the input speech, and their semantic connection. For that purpose, we used the question settings used in [16][24].

Table 2. Likert scale questions used in the user study.

| Scale | Statements (Translated from Japanese) |
|---|---|
| Naturalness | Gesture was natural |
| | Gesture were smooth |
| | Gesture was comfortable |
| Time Consistency | Gesture timing was matched to speech |
| | Gesture speed were matched to speech |
| | Gesture pace was matched to speech |
| Semantics | Gesture was matched to speech content |
| | Gesture well described speech content |
| | Gesture helped me understand the content |

We compare the generated motion of the proposed model with different control groups using video clips. The expected result is that the gestures generated by the proposed model are assigned better scores than the baseline model, and are approaching the level of the ground truth for all aspects in the questionnaires. By randomly choosing samples from the test set, we prepared 5 sets for 5 different utterances, in which there are 4 videos within each set (GT, Baseline, and two for the proposed model). The order of the videos is randomized. Participants are required to assign scores (from 1 to 7, 1 represents strongly disagree, 7 represents stronly agree) for each scale of the gesture performed by the avatar defined in table 2 after watching each video. We recruited 33 participants (17 male, 16 female, all native Japanese speakers, average = 38, standard deviation = 9.5 years old) through a cloud sourcing service.The results are shown in Fig. 3.

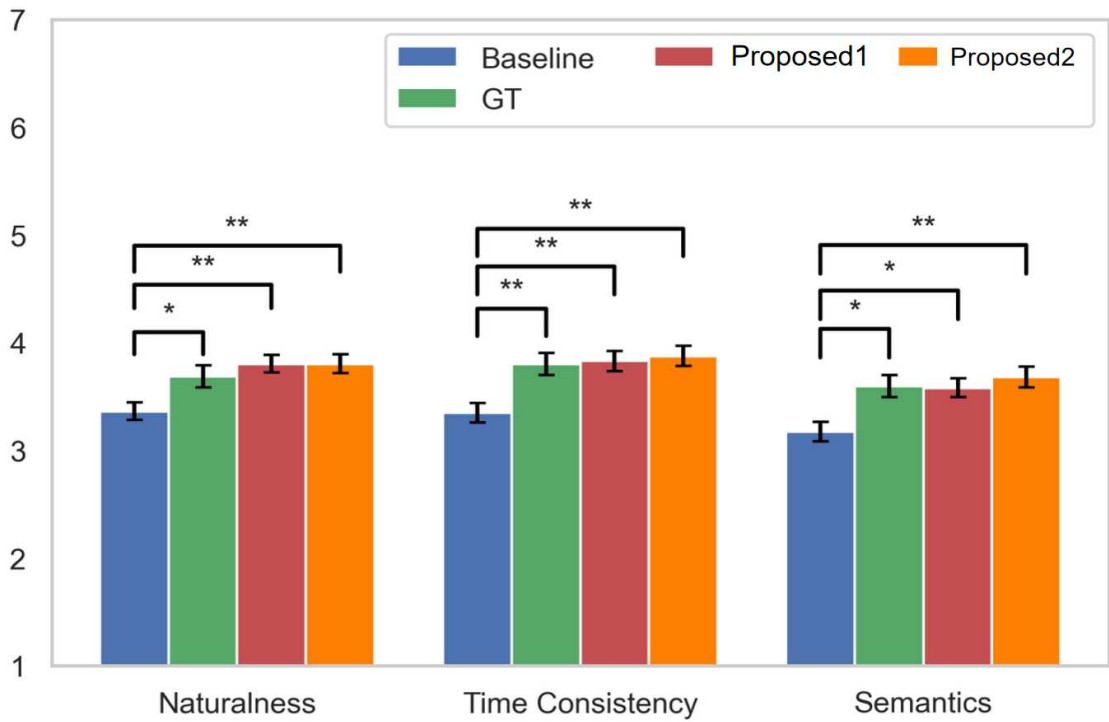

Fig. 3. Subjective evaluation result. Proposed1 and proposed2 are the two synthesized results using the proposed model with different noise vector for the same audio segment input. The error bar represents the standard error of the average value. *: $p < 0.05$, **: $p < 0.01$

Analysis of variance (ANOVA) was conducted to statistically test if there is a significant difference between the scores of the groups in our experiment setting. Scores of all scales passed ANOVA with $p < 0.01$. Tukey's honestly significant difference test (Tukey HSD) was used to test whether there is a significant difference between groups pair-wisely. For the naturalness scale, there was a significant difference between the baseline and ground truth , $p < 0.05$, between baseline and proposed 1 , $p < 0.01$, and between baseline and proposed 2 , $p < 0.01$. There was no significant difference between ground truth and proposed 1, $p = 0.77$, and between ground truth and result 2, $p = 0.77$, and between proposed 1 and proposed 2, $p = 0.9$.

For the time consistency scale, there was a significant difference between baseline and ground truth , $p < 0.01$, and between baseline and proposed 1 , $p < 0.01$, and between baseline and proposed 2 , $p < 0.01$. There was no significant difference between ground truth and proposed 1, $p = 0.9$, between ground truth and result 2, $p = 0.9$, and between proposed 1 and proposed 2, $p = 0.9$.

For the semantics scale, there was a significant difference between baseline and ground truth , $p < 0.01$, between baseline and proposed 1 , $p < 0.05$, and between baseline and proposed 2 , $p < 0.01$. There was no significant difference between ground truth and proposed 1, $p = 0.9$, between ground truth and proposed 2, $p = 0.9$, and between proposed 1 and proposed 2, $p = 0.86$.

### 4.3 Ablation Study

*4.3.1 Continuity loss.* Continuity loss is used in training to force a few frames of motion at the beginning of the following generated sequence to be identical to a few frames of motion at the end of the previous sequence. The expected effect is two-folded. Firstly, it is supposed to be helpful to the consistency of motion between nearby generated sequences, forcing the motion between nearby chunks to be continuous. Secondly, inputting the previous motion sequence can provide context for the generation of the next chunk and maintain the semantics of the generated motion.

We obtained different model parameters with the same training hyper-parameters except for the scale factor for continuity loss. Motions are synthesized using speech input in the test set, by which we observed that without continuity loss in training, the generated motion of the next chunk tends to be independent of the previous motion. That is, even though gestures have at least two phases, preparation and stroke, and stroke is expected to be performed thoroughly before the upcoming gesture, the motion stops before the end of the stroke and starts preparing for the next stroke without continuity loss. Besides, using interpolation solely makes the transition between chunks too smooth and artificial to be human-like. These problems are not visually apparent in the motions synthesized using models that are trained with continuity loss. These observations are all based on the visualization of an avatar in Unity by inputting joint angles produced using models.

In order to study the effect of continuity loss further, synthesized motions are analyzed at a numerical level using KDE. Based on the visual observation, we hypothesized that (1) the log-likelihood of joint angle and position between motions synthesized using models trained with or without continuity loss should not differ too much, because even though the continuity is problematic for motions synthesized using models trained without continuity loss, the poses in these motions are within the bound of the ground truth motion, (2) the log-likelihood of the velocity of joints should be different significantly since the movement under the two settings are visually different. More specifically, the velocity distribution of the two settings should exhibit different patterns, which should also be observed when inspecting the velocity of specific samples.

Experiments are conducted to verify these hypotheses. Joint angle outputs are transformed to joint position using the toolkit in [16], from which the velocity of each joint is calculated for the evaluation of KDE. The results of the KDE evaluation on position and velocity are shown in Table 3, which reveals that continuity loss helps the model produce more similar position and velocity distributions with ground truth. A histogram of the velocity of motions synthesized using models trained with different settings is plotted to understand how unnatural movements are related to the velocity distribution (Fig. 4).

Additionally, these abnormal velocities mainly locate at the transition period of the synthesized motion, which shows that only interpolation is not enough for the continuity between different chunks of motion, as shown in figure 5. Besides, for motions without continuity loss, the velocity always become higher at the beginning and the end of every

Table 3. KDE evaluation results for different groups. Since our model is probabilistic, the results are the average of three times generation and calculation. As a larger log-likelihood value indicates distributions are more similar, all the models trained with continuity loss perform better than those trained without continuity loss, both for position and velocity, which is consistent with our hypotheses. $\lambda_c$ represents the value that used to train the model in equation 1. Numbers inside the brackets are the standard error of KDE calculation. GT represents the beset results that can be approached.

| Group | Log-likelihood of position | Log-likelihood of velocity |
|---|---|---|
| GT | -28.22 (0.01) | 0.62 (0.01) |
| $\lambda_c = 0$ | -49.77 (0.13) | 0.49 (0.01) |
| $\lambda_c = 0.1$ | -48.04 (0.17) | 0.58 (0.01) |
| $\lambda_c = 0.5$ | -49.03 (0.15) | 0.57 (0.11) |
| $\lambda_c = 1$ | -47.79 (0.18) | 0.56 (0.01) |
| $\lambda_c = 2$ | -48.71 (0.11) | 0.57 (0.01) |

motion chunk. Even though it is unclear why this is happening, using continuity loss can help avoid this unexpected behavior.

Based on the observations and experiments above, it is reasonable to conclude that continuity loss helps the model produce continuous and semantically related motions. We leave how to determine the best $lambda_c$ as future work.

## 5 DISCUSSION AND LIMITATIONS

The results of the user study showed that the proposed model can not only yield better results than the baseline model, but also produce natural variations of motions. Additionally, there was no significant difference between the scores of the proposed model and the ground truth, suggesting that the naturalness of the gestures synthesized using the proposed model is approaching the ground truth level.

One main reason that the proposed model outperformed the baseline model is that the synthesized motion using the proposed model has a regular moving rate. On the other hand, the generated motion using the baseline model seems to move too much (i.e., once there are voices, it moves), which is not so natural. Although this can be partially attributed to the fact that the conversion network from coordinates to rotation vector is not optimal, mode collapse, which is a common failure in GAN training, seems to still be dominant for the baseline model. For example, the real data distribution has density on segments with movement and segments without movement for voiced input audio. Due to mode collapse, the baseline model only concentrates its density on part of the real distribution where the arms are moving frequently. As a result, the produced motions are moving too much compared with the real human motions. Instead, in the proposed method, we utilized an advanced training algorithm for WGAN, which reduces the effect of mode collapse, achieving a more human-like moving rate compared with the baseline model. In addition, motion holding (i.e., when the hands/arms are hold in the gesture space during some time) frequently appears in human gestures. Even though we did not explicitly model the frequency or appearance of motion holding in the produced gestures, we observed more motion holding than the baseline model from several synthesized samples in the proposed model. This could be another reason that the proposed model was better evaluated than the baseline model with respect to human perception.

On the other hand, no significant differences were found between the generated motions and the ground truth. However, it is hard to conclude that the generated motions have reached the human level. The reason is that the time of the video in the subjective evaluation is short (around 10 seconds). If the time of video for evaluation becomes longer,

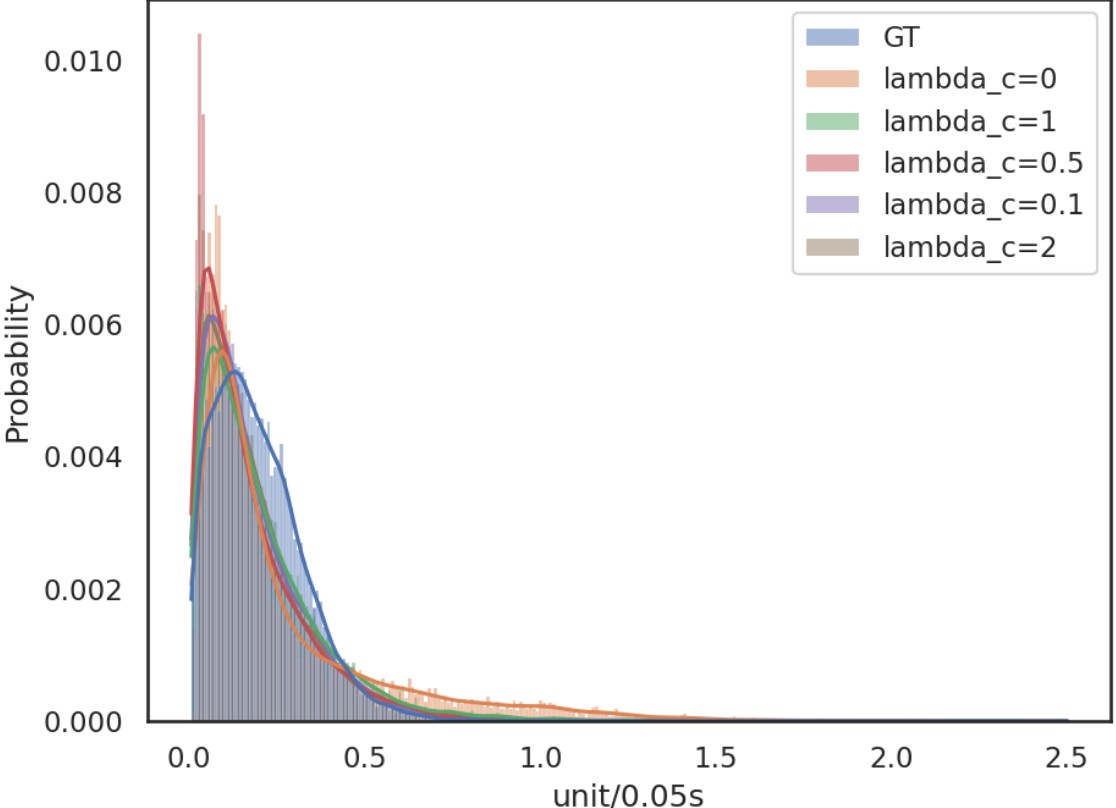

Fig. 4. Average velocity distribution of all joints. While GT represents the ground truth, lambda_c represents the value that used to train the model in equation 1. Lines are fitted densities based on the histrogram. The motion generated using the model trained without continuity loss has more movements with higher velocity than the ground truth, and the motion generated using the model trained with continuity loss has more movements at a lower velocity than the ground truth. Although with continuity loss, the velocity distribution still has space for improvement, it is within the range of the ground truth. However, without continuity loss, the velocity distribution exceeds the range of the ground truth.

e.g. 1 minute, 10 minutes, or even 30 minutes, the evaluation results of the motions generated using the proposed model may be worse than the current result, since the model does not make efforts on learning any long-term dependencies. The effect of changing the length of the video used in the evaluation should be investigated in the future.

Also, although there should be connections between gestures and the context of the speech, these are not included in the current work. The reason is that high-dimensional conditions will drastically reduce the number of samples included in that condition, increasing the difficulty of training. In future work, however, the context in the speech should be included in the model to make the produced gestures more expressive.

## 6  CONCLUSION

In this paper, we proposed a GRU-based model for the generation of gestures from speech. A CNN-based discriminator was designed to train the generator with a WGAN-based learning algorithm in an adversarial manner, leading to better

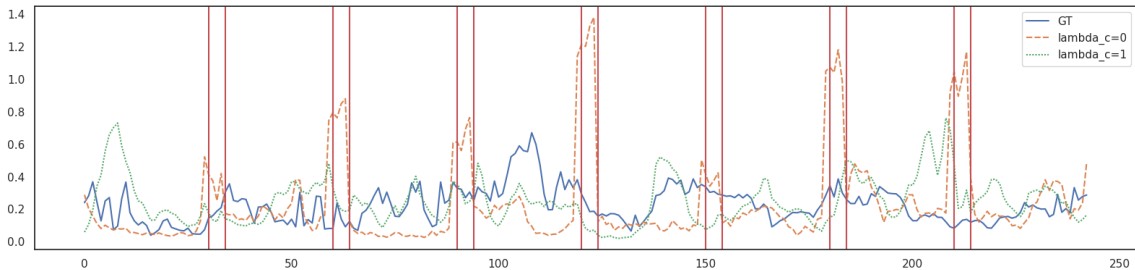

Fig. 5. Velocity evolving by time of a specific sample. While GT represents the ground truth, lambda_c represents the value that used to train the model in equation 1. The X-axis represents time in frame, every frame equals 50 milliseconds. The Y-axis represents the amplitude of the velocity. Periods within red lines are transition parts, i.e., the interpolation between two consecutive motion chunks.

performance for the generator than a state-of-the-art baseline model. We investigated the effects of using a more stable learning algorithm for training GAN on gesture generation, and empirically provided a guideline for researchers who need a system of generating natural gestures from speech. More importantly, the achieved results can be directly applied to agents and robots whose movements are controlled by the joint rotation values, such as the Unity avatar used in this work.

## 7  ACKNOWLEDGMENTS

This work was supported in part by the Grant-in-Aid for Scientific Research on Innovative Areas JP20H05576 (model training), and in part by JST, Moonshot RD under Grant JPMJMS2011 (evaluation experiments).

## 8  APPENDICES

## A  DETAILED MODEL ARCHITECTURE

The detailed model structure and hyper-parameters of the proposed model are provided in Fig. 7 and Fig. 8. The details of the Conv block in the discriminator are shown in Fig. 6. In the figures, FC represents fully-connected layer, and the numbers within brackets indicate input and output sizes of each block. For the 1D-Conv layers of the discriminator, the numbers within brackets indicate input channel, output channel, kernel size, and stride, respectively. For the LeakyReLU blocks, the negative slope is shown within the brackets.

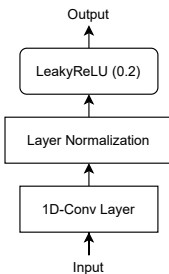

Fig. 6. Discriminator architecture.

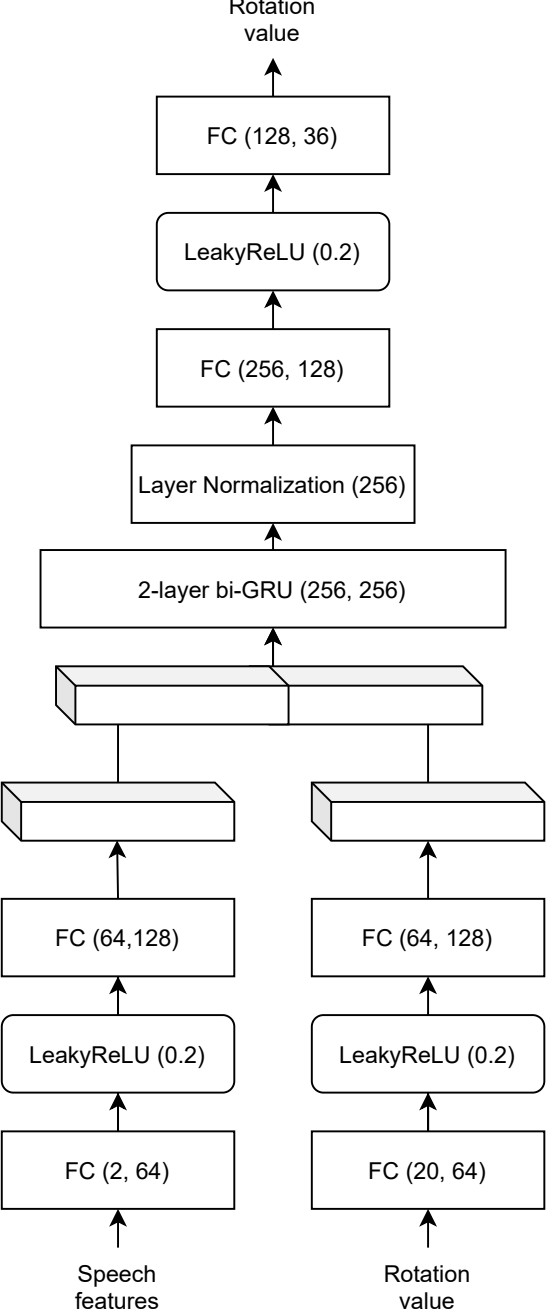

Fig. 7. Generator architecture.

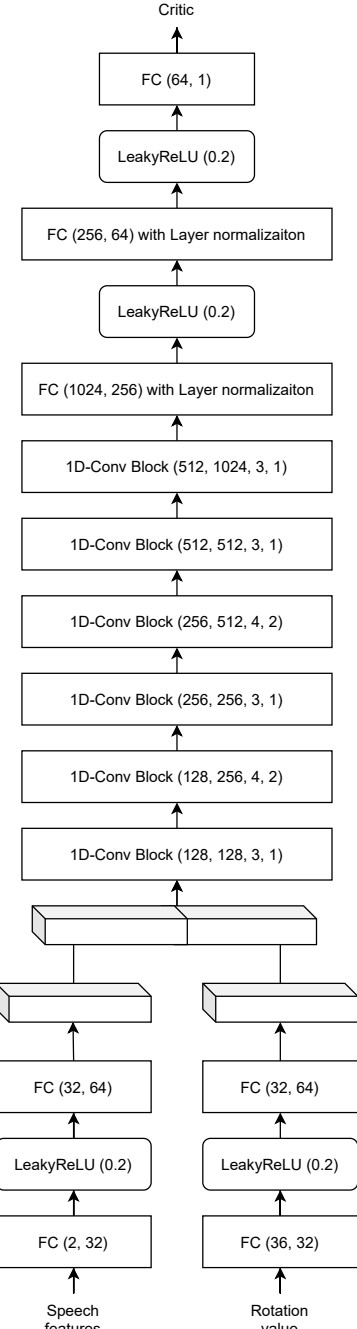

Fig. 8. Discriminator architecture.

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
