# OpenReview forum: "Probabilistic Human-like Gesture Synthesis from Speech using GRU-based WGAN"
_ACM.org/ICMI/2021/Workshop/GENEA — GENEA Workshop 2021 Oral_

### Official Review · Reviewer_SS3V · 2021-07-12
**Probabilistic Human-like Gesture Synthesis from Speech using GRU-basedWGAN**

**Rating:** 6
**Confidence:** 3

**Review:**

The authors propose a new architecture to generate communicative gestures for a virtual agent. They have conducted an objective study and an experimental one to evaluate their model.

The presentation of the model lacks some details.
It is not clear to me why the authors chose to segment the audio into chunks of 1.5sec. Why not relying on IPU Inter-Pausal-Unit?
Communicative gestures have different phases (preparation, hold, stroke...). In particular the stroke is an important phase which is often distinguished by a specific dynamic pattern. I wonder if the continuity loss features, as a second effect, may not leverage the change in dynamism of the gesture (cf comment line 491).
It would have been great to have been able to see videos of the obtained results.
Regarding the subjective evaluation study, I am puzzled of the results. The videos derived from the GT got lower results than the proposed models. How do the authors explain this?
Regarding the questionnaire used in the study, it may be more appropriate not to ask direct questions to participants when one wants to measure a given factor, but rather indirect ones.

At times, the paper is not very easy to read. The authors should shorten some of the lenghty sentences (eg lines 135-140).

---

### Official Review · Reviewer_P2wK · 2021-07-14
**Probabilistic Gesture Generation using WGANs**

**Rating:** 6
**Confidence:** 5

**Review:**

The paper proposes an approach for speech conditioned gesture synthesis that uses WGANs with Gradient penalty over a recurrent GRU layer. A contribution of this paper is to learn to generate a distribution of gestures for the same speech input.

### Strengths
- A potential advantage of this approach is that the generated gestures are sampled from distribution instead of being completely deterministic. The objective metrics leverage the distribution of gestures to measure the log-likelihood of the generated distribution which is more indicative of the correctness of generation as opposed to metrics like MSE.

### Weaknesses
- While the paper describes an approach to generate a distribution of gestures, it does not compare with flow based models which, by design, predict the likelihood of the generated gestures.
- l219: It is not clear why only 1.5 seconds was the window length for training the model. In l234-238, while it is explained how the naturalness of the gestures is maintained, the semantic relatedness of gestures over longer periods of time (I.e some gestures are longer that 1.5 seconds) will not be preserved due to a lack of information from the context. As semantic relatedness is one of the measurement scales in the human study, it might be worth exploring a larger context window.
- Eq 3: Gradient penalty is often applied such that the gradient is not more that _1_. Here the equation seems as if it would encourage the gradient to go to zero which could potentially slow down the training cycles.
- Sec 3.3.4. The dataset has been quickly described in this section. It wasn't clear what dataset was used in this paper. Is it a new dataset? If so, it might be worth stating that as a contribution and describing it in more detail. If it is a pre-existing dataset, a relevant citation would be helpful.
- Eq 4: The model also uses a continuity loss function, which would potentially be useful for shorter samples as used in this paper. But, a couple of concerns
  - Possible Typo: the second term is probably $y_{-k:}^{(i-1)}$ (i.e. last k frames of the (i-1)th sample)
  - why are the last _k_ frames of the _(i-1)_ th sample the same as the first _k_ frames of the _i_ th sample?
  - how is the parameter _k_ chosen?
  - because the loss is dependent on contiguous samples, does the batch only contain contiguous samples? If yes, this could slow down of negatively affect the training process. Randomization of samples is often important for robust training of deep learning models.
- Subjective Evaluation: From what I understand, only 5 videos were used for each model. would 5 videos be enough to measure accurate statistics of the preference scores?

#### Missing References
- l103 - [1] [2]

[1] - Ahuja, Chaitanya, et al. "No Gestures Left Behind: Learning Relationships between Spoken Language and Freeform Gestures." Proceedings of the 2020 Conference on Empirical Methods in Natural Language Processing: Findings. 2020.

[2] - Bhattacharya, Uttaran, et al. "Text2Gestures: A Transformer-Based Network for Generating Emotive Body Gestures for Virtual Agents 2021 IEEE Virtual Reality and 3D User Interfaces (VR). IEEE, 2021.

---

### Official Review · Reviewer_QW6P · 2021-07-15
**A novel and interesting contribution to the workshop**

**Rating:** 7
**Confidence:** 3

**Review:**

This paper proposes a novel extension to GAN-based gesture generation from speech. The work is well presented and I think would make a positive contribution to the workshop. Overall, the level of technical detail is good, and the related work covered is comprehensive to my knowledge. The subjective evaluation is nice to see, but could do with some improvements for the statistical reporting. My detailed suggestions for clarifications and improvements are as follows:

- Section 3.1 - "The audio is segmented into 1.5 seconds with an overlap of 0.2 seconds"
Further details would be helpful for other researchers here. How was this determined? Section 3.2 mentions that long sequences can be broken into short chunks, but still doesn't explain why this time length or overlap setting was chosen.

- Section 3.3.4 - The detail of the motion capture data is quite minimal. The Motion Capture Toolkit is mentioned, but not how many markers were used, the frame rate of the capture, or what the context of the data generation is, i.e., what prompt was given to the person; whether they were a trained actor, or a researcher; giving a presentation, or in an interaction, etc. Including more detail about the dataset would help to provide more context for the subjective evaluation results and for reproduction.

- Section 4 - Why is a filter required on the ground truth? I am wondering specifically what the filter settings are, and whether the source data was particularly noisy, to understand how much smoothing was applied.

- Section 4.2 - How were the subjective video utterances chosen? Were they randomly selected from the ground-truth?

- Section 4.2 - The scale used should be clarified. The paper states that 1 represents negative and 7 represents positive. I am assuming that this was a standard Likert scale though (as suggested by the Table 2 caption), and 1 instead would be 'strongly disagree' and 7 would be 'strongly agree'?

- Section 4.2 - The results of a one-way ANOVA are reported, but several key details are missing:
    - The authors should include details about whether the assumptions of the ANOVA are met, specifically for the homogeneity of variances (e.g., through Levene's test), as this determines whether an ANOVA is an appropriate test.
    - An omnibus test should then be reported for each question (F and p statistics). If the omnibus test is not significant, then the pairwise comparisons should not be conducted.
    - "Scores of all scales passed ANOVA with p < 0.01" - I am not sure what this statement means. If it is stating that all other conditions are significant compared to the baseline, then I think it can be removed. If this is the omnibus test results, then each one should be presented, with the DOF and F statistic.
    - Means and SD's should still be reported for each condition in the pairwise comparisons, not just the p value. Additionally, including an effect size is useful. Without these details, the results of the statistical calculations cannot be reproduced.

- Section 4.2 - the results for the ground-truth are surprising, appearing not to be particularly natural, well timed, or semantically correct. As mentioned above, more dataset detail might help put this in context, but I'm wondering why these scores aren't higher. Maybe this is a result of how the scale is interpreted, or maybe because the 3D model used for the evaluation is not high fidelity? I would be interested to hear the authors thoughts about this, and it could be included as a discussion point.

- Figure 3 - The caption should state what the error bars show; is it standard deviation?

- It would also be very helpful if the authors could include links to videos (I know there is supplemental material with videos, but having a link from the paper would help readers find it), as well as the code, and the dataset used. This would aid in reproduction and is in line with the workshop goals. However, I understand if this is not possible.

---

### Decision · Program_Chairs · 2021-07-19

Accept (Oral)